# Direction of attentional focus in prosthetic training: Current practice and potential for improving motor learning in individuals with lower limb loss

Szu-Ping Lee[1]*, Alexander Bonczyk[1], Maria Katrina Dimapilis[1], Sarah Partridge[1], Samantha Ruiz[1], Lung-Chang Chien[2], Andrew Sawers[3]

1 Department of Physical Therapy, University of Nevada, Las Vegas, Nevada, United States of America,
2 Department of Epidemiology and Biostatistics, University of Nevada, Las Vegas, Nevada, United States of America, 3 Department of Kinesiology and Nutrition, University of Illinois at Chicago, Chicago, Illinois, United States of America

☉ These authors contributed equally to this work.
* szu-ping.lee@unlv.edu

**Data Availability Statement:** All relevant data are within the paper and its Supporting Information files.

## Abstract

### Objective

Adopting an external focus of attention has been shown to benefit motor performance and learning. However, the potential of optimizing attentional focus for improving prosthetic motor skills in lower limb prosthesis (LLP) users has not been examined. In this study, we investigated the frequency and direction of attentional focus embedded in the verbal instructions in a clinical prosthetic training setting.

### Methods

Twenty-one adult LLP users (8 female, 13 male; 85% at K3 level; mean age = 50.5) were recruited from prosthetic clinics in the Southern Nevada region. Verbal interactions between LLP users and their prosthetists (mean experience = 10 years, range = 4–21 years) during prosthetic training were recorded. Recordings were analyzed to categorize the direction of attentional focus embedded in the instructional and feedback statements as internal, external, mixed, or unfocused. We also explored whether LLP users' age, time since amputation, and perceived mobility were associated with the proportion of attentional focus statements they received.

### Results

We recorded a total of 20 training sessions, yielding 904 statements of instruction from 338 minutes of training. Overall, one verbal interaction occurred every 22.1 seconds. Among the statements, 64% were internal, 9% external, 3% mixed, and 25% unfocused. Regression analysis revealed that female, older, and higher functioning LLP users were significantly more likely to receive internally-focused instructions (p = 0.006, 0.035, and 0.024, respectively).

**Funding:** This study was partially funded by the following awards: Lee SP (PI). Motor Learning in Individuals with and at Risk of Lower Limb Loss: Implications for Amputee Rehabilitation. Research Scientist Development Award (1K01HD091449), Eunice Kennedy Shriver National Institute of Child Health and Human Development, National Institute of Health. Lee SP (PI), Hsu YT, Chien LC. Mobility and Patient-Perceived Outcomes of Rehabilitation after Lower Extremity Amputation Surgery. Encompass Health Rehabilitation Research Grant, Encompass Health Corp. The sponsors played no role in the study design, data collection and analysis, decision to publish, or preparation of the manuscript.

**Competing interests:** The authors have declared that no competing interests exist.

## Conclusions

Our results demonstrated that verbal instructions and feedback are frequently provided to LLP users during prosthetic training. Most verbal interactions are focused internally on the LLP users' body movements and not externally on the movement effects.

## Impact statement

While more research is needed to explore how motor learning principles may be applied to improve LLP user outcomes, clinicians should consider adopting the best available scientific evidence during treatment. Overreliance on internally-focused instructions as observed in the current study may hinder prosthetic skill learning.

## Introduction

There are currently more than 2 million people with an amputation living in the United States [1], and approximately 300 to 500 new amputations occur each day [2]. Oftentimes, those who require an amputation have other chronic comorbidities [3]. For example, diabetes is the leading cause of nontraumatic lower limb amputations that accounts for more than half of all amputations in the U.S. [1]. Given the complexity and the chronic nature of causes of amputation, it is important to explore effective rehabilitation strategies for these individuals to address aspects of their disability and to maximize recovery of function after amputation.

Despite the large and increasing number of individuals living with limb loss, current rehabilitation strategies, including prosthetic training, are often inadequate and unstandardized due to a lack of evidence to guide clinical practice [4]. Contemporary research supports the incorporation of motor learning principles to improve motor skill acquisition in clinical settings [5]. Little attention however, has been given to how these concepts can be implemented in prosthetic training [6, 7]. A lack of evidence for effective rehabilitation strategies in regards to post-amputation rehabilitation, and in particular prosthetic skill training, may contribute to suboptimal functional outcomes observed in many lower limb prosthesis (LLP) users despite receiving current standard of care therapy [8].

Empirical evidence from the last two decades have shown how the direction of attentional focus significantly affects motor learning and performance [5, 9]. This theory states that by instructing learners to focus on controlling the motions of body segments or muscle contractions (i.e. internal focus), it may interfere with the natural motor control processes. In contrast, an external focus of attention on the intended movement outcomes allows the learner to adopt a more automatic form of control and is therefore more effective [10–12]. For example, Wulf, McNevin, and Shea (2001) found that performance and learning of a balance task was improved when an external attentional focus was adopted [11]. In their study, the internal focus group was instructed to focus on controlling their feet, while the external focus group was told to focus on controlling the movement of the balance platform they stood on. They found that those in the external focus groups not only performed better during training, they also exhibited better retention of the learned skill. Prosthetic researchers have advocated that the theory of attentional focus may be utilized to improve the effectiveness of prosthetic skill training by instructing the LLP users to focus externally (towards the movement task goals) rather than internally (toward body or prosthetic movements) [6]. The application of this theory, however, remains unexplored within the context of prosthetic rehabilitation. In fact, it

may be more common for prosthetists to use internally-focused instruction and feedback during gait training, as has been demonstrated in physical therapists treating patients with stroke and in other practice fields [13–16].

While the practice of prosthetics typically centers around designing, fabricating, and fitting prostheses, it often also includes training the users on how to properly use their prosthetic device for motor activities such as walking. The training process can be challenging to adult patients with limb loss because it involves controlling the residual limb-prosthesis interface with altered sensory input and motor output [17]. Such challenge often leads to slower learning as well as increased risk of falls and other injuries during the initial phases of learning to use a LLP. Furthermore, a LLP user may have to relearn and re-adapt when different prosthetic components are introduced (e.g. socket design, prosthetic joints, etc.), or when physical changes occur after surgical revision and atrophy of residual limb muscles over time [18, 19]. Because of this emphasis on skill learning, practice, and motor adaptation, it is generally agreed that motor learning strategies including the adoption of an external focus of attention when delivering instructions or feedback to LLP users could benefit prosthetic training, device acceptance, and improve rehabilitation outcomes after amputation [7, 20]. It may be beneficial for prosthetists and other post-amputation care providers to recognize that the words used in their instructions pertaining to how to operate a prosthetic device can have a tangible impact on the patient's learning, as has been demonstrated in other disciplines of rehabilitation [21–24].

The purpose of this study was to evaluate the verbal interaction between prosthetists and LLP users during prosthetic training. Specifically, we sought to compare the frequency and direction of attentional focus (internal, external, and mixed focus) embedded in the verbal instructions and feedback interactions between prosthetists and LLP users in clinical practice settings. We hypothesized that during prosthetic training, the majority of the verbal instructions would direct LLP users' attention internally. A secondary purpose of the study was to determine whether the direction of attentional focus embedded in instructions received by the LLP users was related to the treating prosthetists' experience and/or LLP users' characteristics such as age, sex, time since amputation, and physical function.

## Methods

### Participants

Inclusion criteria for LLP user participants included lower limb amputation involving at least one major joint (i.e. ankle, ankle & knee, or above), 18 years of age or older, and current or planned use of a prosthesis. LLP users were recruited from local prosthetic clinics in the Southern Nevada region. Prosthetists had to be certified, actively practicing prosthetists. These criteria were selected to obtain a convenient sample of typical adult LLP users and practicing prosthetists. Neither the LLP users nor the prosthetists were informed about the purpose of the study to eliminate the risk of possibly affecting the interactions. The sample size of 20 LLP users-prosthetist couples was determined based on a previous study examining similar clinical practice behavior [13].

### Procedure

After a LLP user was recruited, a researcher asked for his/her permission as well as permission from the treating prosthetist to observe and record the session that involved gait and mobility training with their prostheses. Pre-prosthetic sessions (i.e. sessions before a working prosthesis was available to the LLP user) were excluded from this study. The researcher then explained to the LLP user and the prosthetist that they would be recorded during the training session

without any interference or comments. An informed consent form approved by the University of Nevada, Las Vegas Institutional Review Board (IRB) for Biomedical Research was given to each participant to be read and signed prior to data collection.

A smart phone with an audio/video recorder was used for data collection. An external shot-gun microphone was connected to the phone to improve audio quality, particularly to help clearly record the verbal exchanges between the prosthetist and LLP user. If the entire session was in one room, the recorder was fixed to a tripod and placed in a corner of the room to minimize intrusions during the prosthetic training. If the prosthetist and LLP user changed locations, the researcher held the recorder and walked behind to minimize disruption.

After obtaining consent, a data collection sheet was used to collect each LLP user's demographics and medical history, and the treating prosthetist's years of experience. The LLP user questionnaire was used to record ethnicity, gender, age, date, cause, and level of amputation, use of assistive devices, and MFCL-level (provided by the treating prosthetist). Perceived mobility of LLP users was assessed using the Prosthetic Limb Users Survey of Mobility (PLUS-M) 12-item Short Form [25].

## Data analysis

The recorded videos were reviewed and transcribed by one of the two data analysts with the assistance of online transcription services (Rev.com or Otter.ai.). A data analyst cross-checked then analyzed each transcript to identify the frequency of internally focused, externally focused, mixed focus, and unfocused statements embedded in the verbal interactions between the prosthetist and LLP user during each session. These themes were established based on previous research of attentional focus in clinical rehabilitation practice [14]. The thematic analysis procedure was based on the technique described by Pope et al. (2000) which suggested five stages to qualitatively analyze health care interview data (familiarization, identifying a thematic framework, indexing, charting, and interpretation) [26]. In this study, *familiarization* involved re-watching the recordings and taking note of recurring themes. *Identification of a thematic framework* was done by identifying key concepts that could be observed in each LLP user-prosthetist pairing. Subsequently, *indexing* included examining the concept of interest embedded in the recordings (i.e. direction of attentional focus). Summaries of the findings were then arranged in a *chart* to *interpret* themes from the data.

Statements were identified as having an internal focus if they directed a LLP user's attention to one or more body parts, such as their foot, leg, and/or knee. A distinction was made between internally focused comments that were directed towards a LLP user's body, residual limb, intact limb, and/or those directed towards their prosthesis or prosthetic components. Statements were identified as having an external focus if they directed a LLP user's attention towards the desired effects of the movement, such as instructing them to walk towards a target or push off against the ground. Statements that included more than one type of focus (mixed focus) and without a specific focus (unfocused) were categorized separately (Table 1).

An analysis matrix was created to outline the different types of tasks, instructions, and feedback given during each recorded training session (see S1 Appendix for an example of an annotated matrix). The matrix was used to quantify the frequency of usage of the five attentional focus types. Because the recordings were of different lengths and contained different numbers of statements, we computed the percentages of the four types of attentional focus statements relative to the total number of statements in each recording. A similar methodology has been used in previous research studies and was shown to be reliable [14, 27]. A reliability study was conducted based on the first five videos collected. The inter-rater reliability of determining the statement types was examined using 2-way random intraclass correlation coefficient models

**Table 1. Definitions and examples of the attentional focus themes.**

**Definition of an instruction: statements regarding how an action is to be performed**
**Definition of feedback: statements regarding an action in order to encourage, discourage, or modify it. This can be given during or after the action.**

| Theme | Example (#: LLP user participant ID) |
|---|---|
| **Internal focus statement-prosthesis**<br>A statement focusing on movement of the learner's prosthetic body part | **Instruction:**<br>"Rotate your foot out." (#1)<br>"Roll off the toe." (#1)<br>**Feedback:**<br>"The alignment [of the prosthesis] looks good." (#5)<br>"You're swingin' that [referring to prosthesis] really good." (#9)<br>"Yeah, I can see that your heel is off." (#9) |
| **Internal focus statement-intact body**<br>A statement focusing on movement of the learner's intact body part in space. | **Instruction:**<br>"I'm going to have you bend your knee for me." (#3)<br>"Shift your body weight onto this leg." (#3)<br>**Feedback:**<br>"Your hips look more level right there." (#5)<br>"Your skin hangs right over the edge." (#8) |
| **External focus statement**<br>A statement that directs the learner's attention towards the desired effects of the movement. | **Instruction:**<br>"Move forward." (#1)<br>"Try to look at that picture that's in front of us and just slow down your walk." (#10)<br>"Walk towards me and then back towards the wall." (#15)<br>**Feedback:**<br>"You are pretty much there (referring to a target), you might just have to take a couple of steps." (#14) |
| **Mixed focus statement**<br>A statement that includes both internal and external focus. | **Instruction:**<br>"I'm gonna have you step on some of these [referring to wooden blocks]." (#4)<br>"Try lifting up out of it (a foot placement) a little bit and rotating your foot out." (#12)<br>"Come here really quick. I just want to do one tweak with your foot." (#15)<br>**Feedback:**<br>"Is your limb hitting the bottom at all right now?" (#8)<br>"So what I ended up doing was widening your base of support a little bit, and then I also tweaked your foot out just a hair so you're not on the outside as much." (#15) |
| **Unfocused statement**<br>A statement not giving technical instruction or offering encouragement to the learner only. | **Instruction:**<br>"Let's test it out." (P5)<br>"Let's see you walk real quick if you don't mind." (#9)<br>**Feedback:**<br>"Good." (#2)<br>"You did very good." (#3) |

(ICC$_{2,1}$) for absolute agreement based on independent analysis results from the two analysts [28, 29]. The analysts demonstrated excellent inter-rater reliability for classifying the attentional focus themes of interest (ICC = 0.939 and 0.996 for external and internal categories, respectively).

To compare the percentages of the four types of attentional focus statements delivered by the participating prosthetists to the LLP users, the mean percentage value and 95% confidence interval (95% CI) were computed for each focus type. Normality and homogeneity of variance assumptions were assessed using the Shapiro-Wilks test and White test, respectively. Due to the violation of normality assumption, the nonparametric Kruskal-Wallis tests were used to determine whether there were significant differences within the six participating prosthetists regarding the percentages of attentional focus statement types delivered. This analysis was

conducted to examine if the attentional focus usage pattern can be generalized among the participating clinicians with different clinical training and experiential background.

Multi-variate regression analysis using the Tobit model was conducted to determine whether the prosthetists' experience and/or LLP users' characteristics (sex, age, years since amputation, cause of amputation, amputation level, and mobility measured by PLUS-M) were associated with greater use of internal focus instructions and feedback during prosthetic training. The Tobit model was chosen because the dependent variable (i.e. proportion of internal focus statements) was a percentage value bounded by an upper limit of 100% [30]. The estimated coefficients in the Tobit regression model can be interpreted similarly to those in a linear regression model, albeit the association is not on the observed proportions but the uncensored latent variable values [31]. The variance inflation factor was calculated for each predictor to examine multicollinearity of the model. All data analyses were performed using SAS v9.4 (SAS Institute, Cary, North Carolina, USA). The significance level was set to 0.05.

## Results

### Demographics of LLP users and prosthetists

Six prosthetists (2 female, 4 male) from 3 different prosthetic clinics in the Southern Nevada region participated and recruited their LLP user patients for this study. The participating prosthetists' years of clinical experience ranged from 4 to 21 years (mean = 10.0 years, SD = 6.2). Five of the prosthetists were certified by the American Board for Certification (ABC) and one by the Board of Certification (BOC), and four received master's level training in prosthetics and orthotics.

A total of 21 prosthetic training sessions from 21 different LLP users were recorded and analyzed. One participant's video had no sound (#16) so it was excluded from the analysis. The remaining 20 participants consisted of 12 males and 8 females (mean age = 50.2 years, SD = 11.6). Time since amputation ranged from 0.4 years to 27.3 years. PLUS-M T-scores ranged from 21.8 to 71.4 (mean = 47.7, SD = 12.6) indicating a wide range of perceived mobility among the LLP user participants. Causes of amputation varied but the most prevalent was diabetes (n = 7). Table 2 summarizes the characteristics of the LLP users and prosthetists.

### Direction of attentional focus embedded in prosthetic training instructions and feedback

Length of the training sessions ranged from 6 to 32 minutes for a total of 338 minutes over 20 recorded sessions. Fig 1 shows the relative frequency of attentional focus types in each training session. Nine hundred and four individual statements were transcribed and classified. On average, one verbal instruction/feedback was delivered every 22.1 seconds. Of all statements collected, 48% (436/904) were classified as internal focus of attention directed at the prosthesis or prosthetic component, and 15% (138/904) were classified as internal focus of attention directed at the LLP users' body, residual limb, and intact limb. Overall, internally focused statements accounted for 64% of all verbal instructions and feedback (95% CI = 60–67%, range = 44–88%). Furthermore, 9% (77/904, 95% CI = 7–10%) of the statements were classified as external focus of attention, 3% (30/904, 95% CI = 2–4%) were classified as mixed focus of attention, and 25% (223/904, 95% CI = 22–27%) were classified as unfocused. Given that the 95% CI of the internal focus proportion was significantly higher and did not overlap with that of any other focus types, we concluded that the participating prosthetists predominantly used instruction/feedback that invoked an internal focus of attention, particularly for directing LLP

**Table 2. Characteristics of the LLP users and treating prosthetists.**

| LLP user ID | Prosthetist years of experience (ID) | LLP user characteristics | | | | | | | | |
|---|---|---|---|---|---|---|---|---|---|---|
| | | Sex | Age (y) | Years Since Amputation | Cause of Amputation | Amputated Side | Amputation Level | K-Level | PLUS-M | Session time (s) |
| 1 | 11 (A) | F | 34 | 26.6 | Congenital PFFD | L | AK | K3 | 67.1 | 441 |
| 2 | 4 (B) | M | 55 | 1.6 | Diabetes | L | BK | K3 | 49.1 | 860 |
| 3 | 4 (B) | F | 69 | 0.7 | Diabetes | L | BK | K2 | 27.2 | 947 |
| 4 | 4 (B) | F | 54 | 2.1 | Diabetes related necrotizing fasciitis | R | BK | K3 | 49.8 | 1914 |
| 5 | 4 (B) | M | 60 | 1.9 | Diabetes | R | BK | K3 | 57.3 | 1564 |
| 6 | 4 (B) | M | 50 | 5.1 | Osteomyelitis | L | BK | K3 | 49.8 | 871 |
| 7 | 4 (B) | M | 50 | R: 4.3 L: 1.5 | Diabetes | B | BK | K3 | 48.4 | 1125 |
| 8 | 21 (C) | M | 46 | 15.5 | Motorcycle accident | L | AK | K4 | 71.4 | 512 |
| 9 | 12 (D) | M | 51 | 0.6 | Bone infection | R | Knee disarticulation | K3 | 34.1 | 1168 |
| 10 | 11 (A) | M | 51 | 0.8 | Blood clot | R | AK | K3 | 48.4 | 373 |
| 11 | 4 (B) | F | 54 | 2.1 | Diabetes related necrotizing fasciitis | R | BK | K3 | 49.8 | 1664 |
| 12 | 11 (A) | M | 51 | 0.8 | Blood clot | R | AK | K3 | 48.4 | 828 |
| 13 | 11 (A) | F | 22 | 0.4 | Cancer—synovial sarcoma | L | BK | K3 | 41.5 | 616 |
| 14 | 4 (B) | F | 63 | 1.4 | Peripheral artery disease | L | AK | K3 | 39.0 | 1413 |
| 15 | 11 (A) | M | 51 | 0.8 | Blood clot | R | AK | K3 | 48.4 | 494 |
| 16 | | Excluded due to recording device malfunction | | | | | | | | |
| 17 | 12 (D) | M | 51 | 0.6 | Bone infection | R | Knee disarticulation | K3 | 34.1 | 926 |
| 18 | 11 (A) | F | 22 | 0.4 | Cancer—synovial sarcoma | L | BK | K3 | 41.5 | 1943 |
| 19 | 21 (C) | M | 51 | 7.8 | Diabetic ulcer led to bone infection | L | BK | K3 | 64.5 | 1164 |
| 20 | 6 (E) | M | 62 | 27.3 | Train accident | R | BK | K3 | 62.5 | 672 |
| 21 | 6 (F) | F | 56 | 2 | Sores on bottom of foot that would not heal | R | BK | K2 | 21.8 | 809 |

Note: The PLUS-M T-score was a normalized outcome measure used to assess functional mobility of prosthetic limb users where 21.8 indicated the lowest and 71.4 indicated the highest mobility level. T-score of 50.1 represents the 50th percentile [25]. AK = above-the-knee, BK = below-the-knee, F = female, M = male, L = left, R = right, B = bilateral.

users' attention to their prosthetic devices. There were no statistical differences within the six prosthetists regarding the types of attentional focus statements used (p = 0.330–0.945).

We diagnosed the normality (p-value = 0.2655) and variance homogeneity assumptions (p-value = 0.6838), showing that neither one was violated in the regression analysis. The regression model was also free from the multi-collinearity problem, where all variance inflation factors were less than 10 (range = 1.34 to 3.77) in the seven included predictors. The multi-variate regression analysis showed that LLP users' sex, age, and PLUS-M T-scores were significantly associated with the percentage of internal focus instructions and feedback they received (p = 0.006, 0.04, and 0.02, respectively; Table 3). Specifically, female LLP users were more likely to receive internal focus instructions than males (p = 0.006). LLPs with higher PLUS-M T-scores also tend to receive a larger proportion of internally-focused instructions and feedback (a one-point increase in PLUS-M T-score coincided with 0.7% increase in the internal focus instructions received [p = 0.02; Fig 2]). Prosthetists' years of experience were not significantly associated with the proportion of internally-focused language they used (p = 0.42). Years since

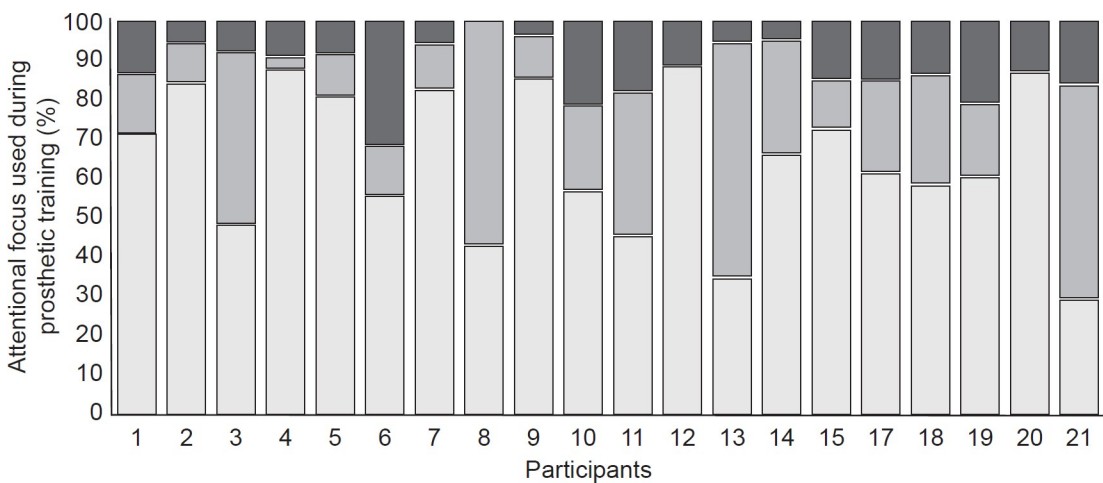

**Fig 1. Distribution of external and internal focus instruction and feedback during prosthetic training (dark gray = % external focus, medium gray = % internal focus on intact body, and light gray = % internal focus on prosthesis).**

amputation, cause of amputation, and amputation level were also not significantly associated with the percentage of internal focus statements received during prosthetic training (p = 0.47, 0.08, and 0.31, respectively; Table 3).

## Discussion

Despite previous research suggesting that incorporating motor learning principles could improve prosthetic training and post-amputation outcomes [6], this is the first study to examine the direction of attentional focus embedded in prosthetic training instructions during clinical practice. Our results demonstrated that the use of instructions and feedback was frequent and ubiquitous during prosthetic training. Our hypothesis was confirmed that most of the verbal interactions delivered by prosthetists to LLP users were focused internally on the movements of the patients' body and/or prosthesis, rather than externally on the intended movement effects.

The first aim of our study was to examine the frequency and direction of attentional focus embedded in the verbal interactions between prosthetists and patients during daily practice. Empirical evidence has accumulated during the last two decades regarding the benefits of

**Table 3. Results of the multivariate analysis of the associations between participant characteristics and percentage of internal focus statements received during prosthetic training.**

| Variable | | Estimate | 95% Confidence interval | | P-value |
|---|---|---|---|---|---|
| Prosthetist experience | | -0.45 | -1.52 | 0.63 | 0.4153 |
| Sex | Male | -16.05 | -27.57 | -4.54 | 0.0063 |
| | Female | | Reference | | |
| Age | | 0.63 | 0.04 | 1.21 | 0.0352 |
| Years since amputation | | -0.36 | -1.33 | 0.62 | 0.4708 |
| Cause of amputation | Dysvascular | -12.13 | -26.10 | 1.84 | 0.0887 |
| | Non-dysvascular | | Reference | | |
| Amputation level | Below the knee | -4.99 | -14.54 | 4.56 | 0.3058 |
| | Above the knee | | Reference | | |
| PLUS-M T-score | | 0.70 | 0.09 | 1.30 | 0.0242 |

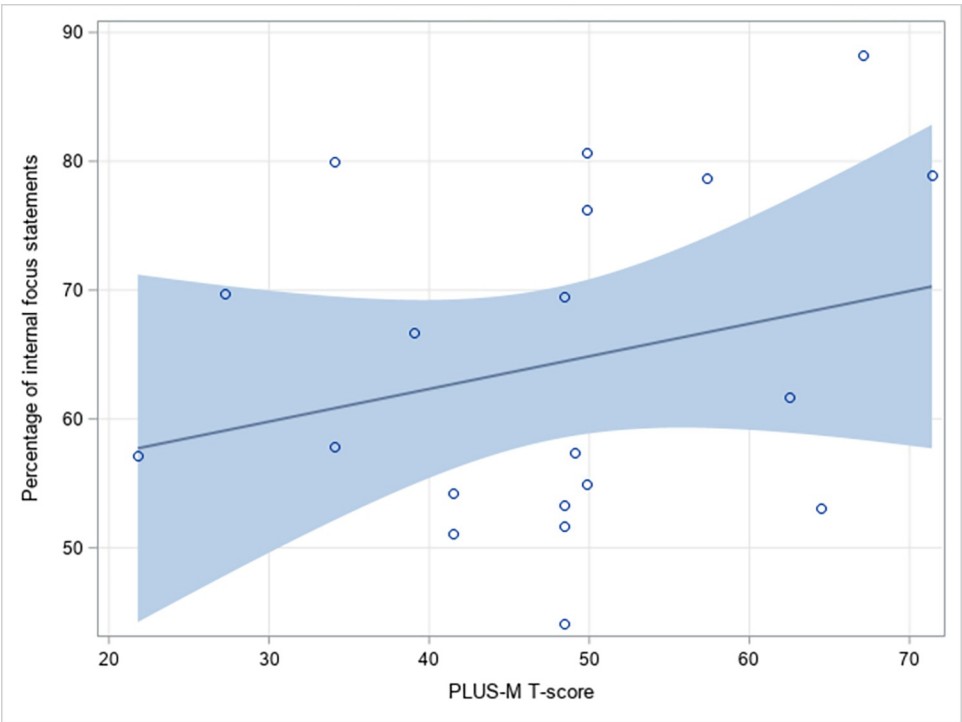

**Fig 2. Relationship of the percentages of internal focus statements versus PLUS-M T-scores.**

adopting an external focus of attention on motor performance and learning as compared to an internal focus [9]. Specifically, previous studies have consistently demonstrated that adopting an external focus before or during the execution of a motor task leads to faster learning, and improved movement effectiveness and neuromuscular efficiency [32–34]. This motor learning principle has profoundly impacted the practice of rehabilitation after neurological injuries such as stroke [5, 13, 14]. In the context of post-amputation rehabilitation, the translation of how external versus internal attentional focus affects learning is perhaps even more pertinent because LLP users need to learn to master the use an external device (i.e. a prosthesis) in place of the lost biological limb.

Our results showed a significant but perhaps unsurprising trend that most (i.e. 64%) of verbal instructions and feedback during prosthetic training were directed internally to the movements of prostheses-users' body and prosthesis. This is comparable to what has been observed in physical therapy for rehabilitation of patients with neurological conditions [13, 14]. For example, Johnson et al. observed eight physical therapy sessions of gait training for patients with stroke and found that 67% of the instructions were internally focused [14]. Previous studies have shown that internally focused instructions can lead to less effective motor learning and performance even when compared to neutral instructions [35, 36]. A possible mechanism underlying motor performance degradation associated with internal focus was the "self-invoking trigger hypothesis" which proposed that frequent evaluation of one's own movements associated with a task (i.e. self-schema) negatively impacts task learning and performance [35]. An external focus that removes the emphasis of controlling the complex coordination of body movements may activate the more natural "automatic" processes of goal-action coupling and promote task automaticity, which is paramount when learning to walk with a prosthesis [11]. As observed in this study of contemporary clinical practice, prosthetists often use internally-

focused language to evaluate a LLP user's movement or to draw attention to specific errors. Such practice, while well-intended, may hinder LLP users' performance and learning.

In speculation that LLP users with certain characteristics may be biased to receive a greater proportion of the presumably less effective internally focused instruction and feedback, our secondary analysis examined these associations. We found that LLP users' sex, age, and perceived mobility were significantly predictive of the percentage of internally-focused instructions and feedback they received. Specifically, women and older LLP users were more likely to receive a higher percentage of internal focus instructions. To the best of our knowledge, no other observational studies to date have reported similar sex and age biases regarding the use of attentional focus in clinical practice. Nevertheless, these biases may potentially impact the training and outcomes of patients, and should be examined further in the future.

Our findings also showed that LLP users with higher PLUS-M T-scores tend to receive a greater proportion of internal focus instructions. It is possible that when treating higher functioning LLP users, prosthetists may choose to focus on addressing subtle body or prosthetic component movements such as symmetry and other specific motor patterns. For example, many prosthetic knee components were designed with the expectation that the user would adopt a pronounced heel-to-toe weight-shifting pattern during the stance phase of gait [37–39], particularly during faster walking to allow rapid knee flexion and greater ground clearance [40, 41]. Internally-focused instructions and feedback may be easier and more intuitive to use when teaching these intricate skills to LLP users, leading to its more prevailing use in higher functioning patients. In a previous study by Kal et al. investigating the association between therapists' attentional focus use in patients with stroke, they found that patients with a longer length of hospital stay tend to receive more external focus instruction and feedback [13]. The authors suggested that the increase in the use of external focus may be a natural progression over the course of rehabilitation to fit the needs, goals, and functioning level of the patients.

Attaining high proficiency in using a prosthesis is important for recovering function after amputation. Prosthetic skill learning may be of even greater importance today owing to the proliferation of sophisticated prosthetic components that are designed and optimized to promote specific and complex movement patterns [42, 43]. While these contemporary prosthetic components facilitate higher levels of functioning, they also demand greater skill and may require additional and more elaborate learning and practice [44]. Compounding the complexity of the learning problem is the varying sensorimotor comorbidities (i.e. reduced proprioception) associated with dysvascular causes of lower limb amputations, which are the current leading reason for acquired amputations. The diminished joint and muscle proprioception may render the traditional modes of feedback about movement quality ineffective (i.e. internal focus instructions emphasizing motion of body segments). For these reasons, we believe that principles of motor learning, in this case the adoption of appropriate external attentional focus associated with prosthetic skill learning, is one of the promising approaches to facilitate rehabilitation of function after amputation.

In light of our findings and the motor learning knowledge regarding the potential benefits of external focus, we propose that prosthetic practitioners (i.e. prosthetists, physical therapists, occupational therapists, and physiatrists) should be mindful about their choice of words when training LLP users on how to perform movement tasks with their prosthesis. As an example, we observed many cases where the prosthetist instructed LLP users to bend or straighten their knees and to kick the feet out and contact the ground with heels during walking. We understand that it is difficult to instruct movements completely without referring to the patient's body or prosthesis, however improved outcomes may be attained if the practitioner can use instructions such as "*walk like you would crush a bug with the heel of your shoes with every step*" (to promote a heel first gait pattern) or using external targets such as markers placed on

the floor to promote larger steps and faster gait [45]. It may be beneficial for prosthetic component manufacturers to develop evidence-guided instructional videos aimed at addressing common movement errors and promoting movement patterns that maximize the performance of the prosthesis and the patient [6]. Further research is needed to identify specific movement goals that are most prevalent in prosthetic training, which can guide the development of motor learning guidelines for improving patient outcomes during prosthetic rehabilitation.

## Limitations

Although 338 minutes of video data were collected and analyzed systematically, a limitation of the present study was that only 20 LLP users and 6 prosthetists from 3 clinics in Southern Nevada region were included. Therefore, the results may be influenced by regional practice trends. Larger scale studies involving a wider range of practice regions and countries are needed to improve generalizability. A second limitation was the presence of a researcher and an audio and video recorder in the room during each session. Even though the prosthetists and LLP user participants were unaware of the purpose of the study, the presence of an observer in the room could have potentially impacted their choice of words and actions. Other factors that can affect instruction and feedback, such as LLP user's level of education, prosthesis use proficiency, and other psychological factors, were not examined in the current study.

## Clinical relevance

While experimental research into the benefits of an external focus of attention during prosthetic skill training is pending, evidence from other clinical models have shown that adopting an external focus can enhance motor performance and benefit long-term learning. Clinicians should adopt the best available scientific evidence of motor learning when treating individuals with lower limb amputation. Overreliance on internally focused instructions may interrupt goal-action coupling and hinder prosthetic skill learning in individuals with limb loss.

## Supporting information

**S1 Data.**
(SAV)

**S1 Appendix. Example of a completed analysis matrix: Focus of attention.**
(DOCX)

## Acknowledgments

We would like to thank the participating LLP users and prosthetists from Prosthetic Center of Excellence, POP Prosthetics, Nevada Orthotics and Prosthetics for accommodating the data collection of this study, particularly during the COVID-19 pandemic in 2020.

## Author Contributions

**Conceptualization:** Szu-Ping Lee, Maria Katrina Dimapilis, Andrew Sawers.

**Data curation:** Szu-Ping Lee, Alexander Bonczyk, Maria Katrina Dimapilis, Sarah Partridge, Samantha Ruiz.

**Formal analysis:** Szu-Ping Lee, Alexander Bonczyk, Maria Katrina Dimapilis, Sarah Partridge, Samantha Ruiz, Lung-Chang Chien.

**Funding acquisition:** Szu-Ping Lee.

**Investigation:** Szu-Ping Lee, Alexander Bonczyk, Sarah Partridge, Samantha Ruiz.

**Methodology:** Szu-Ping Lee, Alexander Bonczyk, Maria Katrina Dimapilis, Sarah Partridge, Samantha Ruiz, Lung-Chang Chien.

**Project administration:** Szu-Ping Lee.

**Software:** Lung-Chang Chien.

**Supervision:** Szu-Ping Lee.

**Validation:** Szu-Ping Lee, Lung-Chang Chien, Andrew Sawers.

**Visualization:** Szu-Ping Lee, Lung-Chang Chien.

**Writing – original draft:** Szu-Ping Lee, Alexander Bonczyk, Maria Katrina Dimapilis, Sarah Partridge, Samantha Ruiz, Lung-Chang Chien.

**Writing – review & editing:** Szu-Ping Lee, Andrew Sawers.

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
