## [Decision Letter · Decision Letter 0]

1 May 2022

PONE-D-22-00709Direction of attentional focus in prosthetic training: current practice and potential for improving motor learning in individuals with lower limb lossPLOS ONE

Dear Dr. Lee,

Thank you for submitting your manuscript to PLOS ONE. After careful consideration, we feel that it has merit but does not fully meet PLOS ONE’s publication criteria as it currently stands. Therefore, we invite you to submit a revised version of the manuscript that addresses the points raised during the review process.

I was able to obtain comments from only one reviewer for your manuscript. However, the comments from this reviewer are very thorough, particularly in relation to the statistical analyses and the discussion section, and I have also read the manuscript carefully myself. I am therefore of the opinion that if you are able to respond appropriately to all the reviewer's comments and points, the manuscript will be considerably strengthened.

We look forward to receiving your revised manuscript.

Kind regards,

Neil R. Harrison

Academic Editor

PLOS ONE

Journal Requirements:

“This study was partially supported by Eunice Kennedy Shriver National Institute of Child Health and Human Development, National Institute of Health (1K01HD091449) and the University of Nevada, Las Vegas Department of Physical Therapy. The funders played no role in the design, conduct, or reporting of this study.”

“This study was partially funded by the following awards:

Lee SP (PI). Motor Learning in Individuals with and at Risk of Lower Limb Loss: Implications for Amputee Rehabilitation. Research Scientist Development Award (1K01HD091449), Eunice Kennedy Shriver National Institute of Child Health and Human Development, National Institute of Health.

Lee SP (PI), Hsu YT, Chien LC. Mobility and Patient-Perceived Outcomes of Rehabilitation after Lower Extremity Amputation Surgery. Encompass Health Rehabilitation Research Grant, Encompass Health Corp.

The sponsors played no role in the study design, data collection and analysis, decision to publish, or preparation of the manuscript.”

Reviewers' comments:

Reviewer's Responses to Questions

**Comments to the Author**

1. Is the manuscript technically sound, and do the data support the conclusions?

Reviewer #1: Yes

2. Has the statistical analysis been performed appropriately and rigorously? 

Reviewer #1: Yes

3. Have the authors made all data underlying the findings in their manuscript fully available?

Reviewer #1: Yes

4. Is the manuscript presented in an intelligible fashion and written in standard English?

Reviewer #1: Yes

5. Review Comments to the Author

Reviewer #1: General comments

This study makes a novel attempt to describe the nature of instructional content delivered by prosthetists during rehabilitation sessions with lower limb prosthetic users. Interestingly, the results indicate a large bias towards verbal instructions that direct the patient’s attention towards the explicit monitoring and control of the body/prosthesis (i.e., internal focus). Such a finding corroborates with other fields of rehabilitation and acts as an extremely important point of reference for research moving forward. Indeed, there is substantial evidence that such instructions can be detrimental to motor control and hinder rehabilitation process. It is hoped that these findings drive future research to tackle this potential flaw in current practices.

In general, I applaud the authors for an extremely well-written paper that adequately highlights the importance of the work and its possible applications. The methodology is extremely thorough and well thought-out, overcoming many limitations that are typically associated with this type of research. I don’t have any huge concerns with the paper, but I do have many points that I would like the authors to address and clarify. In particular, I feel the statistical process could be clearer as I found myself somewhat confused with the components of your regression model and ANOVA. I also feel the authors could reshape their discussion to ensure their key findings are presented prior to the discussion of additional findings that were perhaps an afterthought throughout. The discussion is also somewhat lengthy and repetitive in places, but this is perhaps a preference of my own rather than a necessary change. I recommend several studies that could be used to strengthen the breadth and rationale of this study.

I thoroughly enjoyed reading this work. Thanks!

Minor comments

Introduction

L11:17 – Your opening section does a great job of highlighting the need for evidence-based practice. A recent review paper by Parr et al. (2021) also highlights the importance of applying motor learning principles to prosthesis rehabilitation, including a specific section on attentional focus. Perhaps this review could be added to strengthen this section and highlight recent calls for research such as yours.

• Parr, J. V., Wright, D. J., Uiga, L., Marshall, B., Mohamed, M. O., & Wood, G. (2021). A scoping review of the application of motor learning principles to optimize myoelectric prosthetic hand control. Prosthetics and Orthotics International.

L18:32 - Perhaps this section is a little backwards - it might be clearer to first explain to the naive reader what external vs internal instructions are, before exploring the findings of relevant literature. Knowledge of these terms is a little assumed.

L42:45 - This is an important statement, but I think you need to briefly explain WHY it is agreed. What exactly are the mechanisms underpinning the benefits of an external focus (e.g., Poolton & Maxwell, 2006)? For prosthetics, two recent studies have explored and discussed the benefit of achieving an external focus via “serious gaming” (Kristoffersen et al., 2020, 2021), thus reinforcing the need for application to lower limb control. From a mechanistic point of view, a study in upper-limb prosthetics by Parr et al. (2019) found benefits of adopting an external focus via gaze training and proposed such benefits might be supported by lower cognitive demands, unfreezing the degrees of freedom, and lowering the tendency for engaging in conscious motor processing. These points could strengthen your rationale.

• Poolton, J. M., Maxwell, J. P., Masters, R. S. W., & Raab, M. (2006). Benefits of an external focus of attention: Common coding or conscious processing?. Journal of sports sciences, 24(1), 89-99.

• Parr, J. V. V., Vine, S. J., Wilson, M. R., Harrison, N. R., & Wood, G. (2019). Visual attention, EEG alpha power and T7-Fz connectivity are implicated in prosthetic hand control and can be optimized through gaze training. Journal of neuroengineering and rehabilitation, 16(1), 1-20.

• Kristoffersen, M. B., Franzke, A. W., Van Der Sluis, C. K., Murgia, A., & Bongers, R. M. (2020). Serious gaming to generate separated and consistent EMG patterns in pattern-recognition prosthesis control. Biomedical Signal Processing and Control, 62, 102140.

• Kristoffersen, M. B., Franzke, A. W., Bongers, R. M., Wand, M., Murgia, A., & van der Sluis, C. K. (2021). User training for machine learning controlled upper limb prostheses: a serious game approach. Journal of NeuroEngineering and Rehabilitation, 18(1), 1-15.

L52:53 – Has your hypothesis regarding the frequency of internal focus instructions been justified enough here. Indeed, I believe there is plenty of research available to lead you towards this conclusion that could be used to pre-empt this statement. Two recent studies by Kristoffersen et al. (2020, 2021) have also made this point and might be worth checking.

• Kristoffersen, M. B., Franzke, A. W., Van Der Sluis, C. K., Murgia, A., & Bongers, R. M. (2020). Serious gaming to generate separated and consistent EMG patterns in pattern-recognition prosthesis control. Biomedical Signal Processing and Control, 62, 102140.

• Kristoffersen, M. B., Franzke, A. W., Bongers, R. M., Wand, M., Murgia, A., & van der Sluis, C. K. (2021). User training for machine learning controlled upper limb prostheses: a serious game approach. Journal of NeuroEngineering and Rehabilitation, 18(1), 1-15.

L56:57 – Again, I am not sure this hypothesis has been fully justified. Could the authors elaborate on how they fell upon this? Is there any research you could briefly introduce?

Methods

L74:75 – An interesting exclusion. I wonder how pre-prosthetic instructions affect subsequent prosthesis interaction.

L113:114 – Perhaps the reader would benefit from the other two types of instruction being defined here?

L131:133 - A bit more detail is needed here...

I believe the Kruskal-Wallis is an independent ANOVA? But from what I understand your IV here is "instruction" for which each prosthetist would have 3 values. Would this not require a repeated measures ANOVA of sorts? Am I right in thinking that here you are comparing the 6 prosthetists? If so, then were scores aggregated for a single prosthetist when they have multiple pairings? Or have you actually included 21 prosthetist-patient pairs in this analysis? If so, this is a little tricky because technically your sample isn't independent as the same prosthetist would have contributed to several observations? Some clarification on this process would be appreciated.

L135:140 – Unfortunately I am not too familiar with the Tobit model. However, I found it difficult to understand exactly how your model was structured. From what I understand, you performed three separate regression models with each instruction type (internal, external, mixed) acting as the dependent variable for each model? Prosthetist’s experience and LLP characteristic variables were then used as a selection of predictor variables? If so, I think you can make this clearer to the reader. Also, I believe this process would involve a high number of separate bivariate regressions to be performed. Did the authors do anything to control for the inflated error rate?

L175:177 – Again to clarify, does this mean there was no difference in the frequency of instruction type across the six prosthetists? In other words, your IV is instruction type (internal, external, mixed) for which you have 6 observations each? This seems a little contradictory to the finding of a high bias towards internal instructions reported later. I apologise if I am misunderstanding.

Discussion

General - I might suggest a re-structuring of your discussion. Your key finding (bias towards internal) could be discussed first to highlight its importance. The findings regarding the biases should come later as they are less central to your paper.

L226:227 - These findings are very interesting. However, I'm wondering if the authors provide enough (if any) justification for running this analysis in the first place. It would also be meaningful to elaborate on these findings a little and provide some evidence-based speculation (if possible) to better set-up future investigations into these biases. At the moment, these analyses feel rather throw-away.

L229:231 - Could the authors delve into the rehab/sporting literature to find evidence that might support these claims? I believe there is ample evidence that coaches do indeed tend to refine subtle movement corrections in the search for the “perfect technique”. Maybe this could be used to improve the breadth of these findings?

L252:254 - Not sure these need to be defined again. A clearer definition in introduction as suggested would perhaps suffice by this point.

L286:289 – This is an interesting point and sounds a lot like an analogy! Indeed, such instructions have been applied to the re-learning of gait in Parkinson’s. For example, Jie et al. () told PD patients to pretend they were “following footprints in the sand” which led to rehabilitation gains. The review paper by Parr et al. (2021) covers the application of analogy learning for prosthesis rehabilitation and might be worth checking.

• Jie, L. J., Goodwin, V., Kleynen, M., Braun, S., Nunns, M., & Wilson, M. (2016). Analogy learning in Parkinson's disease: a proof-of-concept study. International Journal of Therapy and Rehabilitation, 23(3), 123-130.

6. PLOS authors have the option to publish the peer review history of their article (what does this mean?). If published, this will include your full peer review and any attached files.

Reviewer #1: No

---

## [Author Response · Author response to Decision Letter 0]

9 May 2022

A detailed point-by-point responses to all reviewer comments in bold in the “Response to reviewer comments” document.

---

## [Decision Letter · Decision Letter 1]

12 Jun 2022

PONE-D-22-00709R1Direction of attentional focus in prosthetic training: current practice and potential for improving motor learning in individuals with lower limb lossPLOS ONE

Dear Dr. Lee,

Thank you for submitting your revised manuscript to PLOS ONE. After careful consideration, we feel that it has merit but does not fully meet PLOS ONE’s publication criteria as it currently stands. Therefore, we invite you to submit a revised version of the manuscript that addresses the points raised during the review process.

You have already made a number of changes which have considerably strengthened the revised manuscript. However, Reviewer 1 has provided a number of additional points below that need to be carefully and satisfactorily addressed before the manuscript can be accepted for publication. 

We look forward to receiving your revised manuscript.

Kind regards,

Neil R. Harrison

Academic Editor

PLOS ONE

Journal Requirements:

Reviewers' comments:

Reviewer's Responses to Questions

**Comments to the Author**

1. If the authors have adequately addressed your comments raised in a previous round of review and you feel that this manuscript is now acceptable for publication, you may indicate that here to bypass the “Comments to the Author” section, enter your conflict of interest statement in the “Confidential to Editor” section, and submit your "Accept" recommendation.

Reviewer #1: (No Response)

2. Is the manuscript technically sound, and do the data support the conclusions?

Reviewer #1: Partly

3. Has the statistical analysis been performed appropriately and rigorously? 

Reviewer #1: No

4. Have the authors made all data underlying the findings in their manuscript fully available?

Reviewer #1: Yes

5. Is the manuscript presented in an intelligible fashion and written in standard English?

Reviewer #1: Yes

6. Review Comments to the Author

Reviewer #1: Response to authors

I thank the authors for making a great attempt to address my concerns. The structure and flow of the paper is much improved. However, there is still a more detail I would like in places, particularly regarding the analyses. I feel some of the responses have somewhat passively addressed my concerns and I worry that there is a misinterpretation of non-significant results. I would therefore appreciate further elaboration upon the following points:

Authors:

“As the focus of this study was observational and on lower limb prosthesis, we decided to put the Kristoffersen et al. studies in the discussion section to avoid overwhelming the readers, particularly clinicians, in the introduction. The following two references were added to line 270-273 in the discussion to highlight the need to understand the science underlying prosthetic training in response to the more sophisticated control requirements of modern prostheses.”

RESPONSE:

My focus here was not primarily to highlight the Kristoffersen paper here per se. Rather, I was hoping the authors could provide some evidence in general to support the prediction that internal focus instructions would be highly frequent during rehabilitation. The authors make a good attempt to explain why internal focus instructions might be less effective, but little to suggest this practice would be common. I again feel as though the rationale for the study and the justification for your hypotheses could be strengthened by literature on the instructional content of rehabilitation practice. For example, is there any subjective data from therapists or patients across any domains that suggests a bias towards an internal focus? Indeed, you actually provide this information in the discussion. Could it be brought forward?

Authors:

“Our primary justification for the study was that instructions to prosthetic device users may impact how well they learn to use the device. Clinical application of this theory has been demonstrated in several rehabilitation studies, which were cited here. To more clearly delineate our intention, we revised this sentence (line 46-49) to the following:”

RESPONSE:

I believe my initial response was slightly too vague and therefore remains somewhat unanswered. To be specific, I want to know why the authors think IF instructions would be utilised less with higher function LLP users? Is there a specific rationale for this hypothesis? Currently it doesn’t seem justified. Are you suggesting that high functioning LLP users are more skilled because they are not receiving If instructions? Or are you suggesting that IF instructions are used more at the early stage of learning and less so when an LLP users becomes more skills? Either of these explanations need justification. Again, these points are better addressed in the discussion and therefore reads as though the hypotheses were derived post-hoc.

Authors:

We provided the definitions and examples for “mixed focus” and “unfocused” statements in Table 1. Specifically, mixed focus was defined as: “A statement that includes both internal and external focus”, and unfocused statement was defined as: “A statement not giving technical instruction or offering encouragement to the learner only.”

RESPONSE:

I was aware of the definitions in Table 1 but found it odd that only one type of instruction was defined in text. Is there a reason for this? Upon first reading I felt as though I would have benefited from all being defined in the same section.

Authors:

Thank you for the suggestion. The Kruskal-Wallis tests were done to compare among the 6

prosthetists to see if there were any significant differences in their uses of internal and external

languages. The percentage for each type of statement used was aggregated for each prosthetist.

RESPONSE:

Unfortunately, I am more confused now! If each type of statement were aggregated for each prosthetist, then you must have three values for each of the six prosthetists, one for IF, EF and mixed? You are then running the ANOVA to determine if there is a main effect of instruction type, and whether the group of six prosthetist tend to use one type of instruction more than another. If this is the case, then your design is technically repeated measures, and the Kruskal-Wallis is not suitable?

Alternatively, I don’t understand how you can run an analyses that compares each prosthetists individually to determine whether one prosthetist might use a type of instruction more than another?

I would need much more clarity here.

Author:

Thank you for the suggestion. The Kruskal-Wallis tests were done to compare among the 6

prosthetists to see if there were any significant differences in their uses of internal and external

languages. The percentage for each type of statement used was aggregated for each prosthetist.

We revised this statement to the following to clarify:

“The nonparametric Kruskal-Wallis test was used to determine whether there was a significant difference in the attentional focus statement types delivered by the six participating prosthetists in the study.”

The revised sentence above now exactly corresponds with the following sentence in the results

section:

“There were no statistical differences in the types of attentional focus statements delivered by

the six prosthetists during the training sessions (p=0.330-0.945).”

RESPONSE:

So to confirm, there was no significant different in the type of instruction delivered? This would suggest that IF instructions were not used any more frequently than EF and mixed instructions? Although the descriptive statistics suggest a bias towards IF instructions you inferential statistics do not? Unless I am mistaken this would prevent you from drawing the conclusion that

“Our hypothesis was confirmed that most of the verbal interactions delivered by prosthetists to LLP users were focused internally on the movements of the patients’ body and/or prosthesis, rather than externally on the intended movement effects.”

and

“Our results showed a significant but perhaps unsurprising trend that most (i.e. 63%)…”

Please can the authors explain this discrepancy?

Authors:

This secondary analysis was done to examine whether certain characteristics of the participants

(clinicians and patients) were predictive of greater use of the presumably less effective internal

focus instruction and feedback. We move this discussion to after the primary finding discussion,

and added the following sentence:

RESPONSE:

I still feel the authors could attempt to speculate their finding regarding their finding that age and gender were related to frequency of IF. This would help better set up future work.

7. PLOS authors have the option to publish the peer review history of their article (what does this mean?). If published, this will include your full peer review and any attached files.

Reviewer #1: No

---

## [Author Response · Author response to Decision Letter 1]

14 Jun 2022

Detailed response to reviewer's comments is provided in a separate file document (Response to reviewer comments v2).

---

## [Decision Letter · Decision Letter 2]

20 Jun 2022

Direction of attentional focus in prosthetic training: current practice and potential for improving motor learning in individuals with lower limb loss

PONE-D-22-00709R2

Dear Dr. Lee,

We’re pleased to inform you that your manuscript has been judged scientifically suitable for publication and will be formally accepted for publication once it meets all outstanding technical requirements.

Kind regards,

Neil R. Harrison

Academic Editor

PLOS ONE

Additional Editor Comments (optional):

Reviewers' comments:

Reviewer's Responses to Questions

**Comments to the Author**

1. If the authors have adequately addressed your comments raised in a previous round of review and you feel that this manuscript is now acceptable for publication, you may indicate that here to bypass the “Comments to the Author” section, enter your conflict of interest statement in the “Confidential to Editor” section, and submit your "Accept" recommendation.

Reviewer #1: All comments have been addressed

2. Is the manuscript technically sound, and do the data support the conclusions?

Reviewer #1: Yes

3. Has the statistical analysis been performed appropriately and rigorously? 

Reviewer #1: Yes

4. Have the authors made all data underlying the findings in their manuscript fully available?

Reviewer #1: Yes

5. Is the manuscript presented in an intelligible fashion and written in standard English?

Reviewer #1: Yes

6. Review Comments to the Author

Reviewer #1: The authors have sufficiently addressed all of my concerns. I applaud the authors on a really cool study with findings that will be extremely informative to the field.

I want to also thank the authors for an enjoyable and considerate review process.

7. PLOS authors have the option to publish the peer review history of their article (what does this mean?). If published, this will include your full peer review and any attached files.

Reviewer #1: No

---

## [Editor Report · Acceptance letter]

27 Jun 2022

PONE-D-22-00709R2 

Direction of attentional focus in prosthetic training:
Current practice and potential for improving motor learning in individuals with lower limb loss 

Dear Dr. Lee:

I'm pleased to inform you that your manuscript has been deemed suitable for publication in PLOS ONE. Congratulations! Your manuscript is now with our production department. 

Kind regards, 

on behalf of

Dr. Neil R. Harrison 

Academic Editor

PLOS ONE